# Formative mixed-method multicase study research to inform the development of a safer sex and healthy relationships intervention in further education (FE) settings: the SaFE Project

Honor Young,[1] Catherine Turney,[1] James White,[1,2] Ruth Lewis,[3] Christopher Bonell[4]

► Pre-publication history and additional material is published online only. To view, please visit the journal online (http://dx.doi.org/10.1136/bmjopen-2018-024692).

[1]Centre for the Development and Evaluation of Complex Interventions for Public Health Improvement (DECIPHer), Cardiff University, Cardiff, UK
[2]Centre for Trials Research, School of Medicine, Cardiff University, Cardiff, UK
[3]Institute of Health and Wellbeing, MRC Social and Public Health Sciences Unit, Glasgow, UK
[4]London School of Hygiene and Tropical Medicine, Public Health and Policy, London, UK

**Correspondence to**
Dr Honor Young;
youngh6@cardiff.ac.uk

## ABSTRACT

**Objectives** Sexual health includes pleasurable, safe, sexual experiences free from coercion, discrimination and violence. In the UK, many young people's experiences fall short of this definition. This study aimed to inform the development of a safer sex and healthy relationships intervention for those aged 16–19 years studying in further education (FE) settings.

**Design** A formative mixed-method multicase study explored if and how to implement four components within a single intervention.

**Setting** Six FE settings in England and Wales and one sexual health charity participated between October and July 2015.

**Participants** Focus groups with 134 FE students and 44 FE staff, and interviews with 11 FE managers and 12 sexual health charity staff, first explored whether four candidate intervention components were acceptable and could have sustained implementation. An e-survey with 2105 students and 163 staff then examined potential uptake and acceptability of components shortlisted in the first stage. Stakeholder consultation was then used to refine the intervention.

**Intervention** Informed by a review of evidence of effective interventions delivered in other settings, four candidate intervention components were identified which could promote safer sex and healthy relationships among those aged 16–19 years: 1) student-led sexual health action groups; 2) on-site sexual health and relationships services; 3) staff safeguarding training about sexual health and relationships and 4) sex and relationships education.

**Results** On-site sexual health and relationships services and staff safeguarding training about sexual health and relationships were key gaps in current FE provision and welcomed by staff, students and health professionals. Sex and relationships education and student-led sexual health action groups were not considered acceptable.

**Conclusions** The SaFE intervention, comprising on-site sexual health and relationships services and staff safeguarding training in FE settings, may have potential promoting sexual health among FE students. Further optimisation and refinement with key stakeholders is required before piloting via cluster randomised controlled trial.

### Strengths and limitations of this study

► This formative mixed-method multicase study research addresses the gap in attention paid to the development phase in intervention research design.
► The research informs the development of a new, universal intervention to improve sexual health and healthy relationships in further education (FE) settings addressing the dearth of interventions in this setting.
► Accessing an accurate sampling frame for the FE population poses challenges and warrants further methodological exploration in future research.
► Two candidate intervention components were identified as important gaps in current FE provision that were acceptable and wanted by FE staff and students, and sexual health professionals.
► These require optimisation and feasibility testing with key stakeholders before piloting via cluster randomised controlled trial.

## INTRODUCTION

Sexual health includes pleasurable, safe sexual experiences free from coercion, discrimination and violence.[1] In the UK, many young people's experiences fall short of this. Of young people in further education (FE), over 50% report experience of dating or relationship violence (DRV).[2] The median age for most recent non-volitional sex (NVS; sex against one's will) is 18 among men and 16 among women.[3] The UK also has the highest rate of under-18 births in Western Europe,[4] 21% of unplanned pregnancies occur among those aged 16–18 years[5] and the 16–24 years age group accounts for over half of chlamydia and gonorrhoea diagnoses.[6] Sexually transmitted infections (STIs), unplanned teenage pregnancies and DRV are associated with adverse medical, social, educational and economic outcomes[7–10] and costs to health and public services. The National Health Service (NHS) costs of STIs and

unintended pregnancies for 2013–2020 are an estimated £11.4 billion, with a further £73 billion for other government departments.[11] In 2008, domestic violence was estimated to cost the UK NHS £1.73 billion per year.[12]

Reducing STIs and unplanned pregnancies among young people is a priority for governments internationally.[13–18] The UK government[19] and WHO[20] have called for new approaches that also address NVS among young people. Systematic reviews suggest that comprehensive interventions combining sexual health knowledge, contraception availability and broader youth development are most effective at improving sexual health outcomes and preventing teenage conceptions.[21–23] Cochrane and Campbell reviews[24 25] and the National Institute for Health and Care Excellence guidance[26] recommend further research on multicomponent interventions which tend to be more cost-effective[27] and are less likely to generate socioeconomic inequalities.[28]

FE settings, akin to technical and FE colleges in Australia and community colleges in the USA, primarily serve the age group of 16–19 years. Socioeconomically diverse and of a broader age range than in university settings, FE provides an optimal setting for intervention work. In England, 1.2 million people aged 16–18 years study in FE settings with increasing participation across all social groups.[29] Heterogeneous settings with a transient student population, FE sites vary considerably in size, number and type of students, and in the range of programmes and services offered. Significant amounts of normalised gender-based harassment and DRV go unchallenged within educational environments, including FE.[2 30] Although there is strong evidence for a comprehensive, 'health promoting schools' approach,[31 32] there is limited evidence on its application for sexual health in FE settings.

The Medical Research Council's guidance for the development and evaluation of complex interventions provides a four-phase framework comprising: development, feasibility and piloting, evaluation and implementation.[33] The first phase involves the development of an intervention's theoretical rationale, inputs, processes and mechanisms of change; identifying underpinning 'active ingredients' and how intervention components are expected to interact with each other and the context of delivery to generate outcomes.[34] Little attention is however paid to this developmental phase.[35] Fletcher *et al*[35] advocate the use of multicase study research to support intervention development and modelling, by increasing understanding of the socioecological context, exploring potential intervention delivery and hypothesising mechanisms of action.[35]

The SaFE Project aimed to identify intervention components to promote safer sex and healthy relationships in FE settings which were acceptable, perceived to be a priority for students and FE staff and could be implemented sustainably.

Informed by a review of evidence of effective interventions delivered in other settings, four components were identified which could promote safer sex and healthy relationships among those aged 16–19 years. An initial theory of change (online supplementary appendix 1) constructed using these components described mechanisms, whereby student-led sexual health action groups led to the restructuring of institutional environments to reduce sexual harassment and risk behaviours[36 37]; accessible, youth-friendly sexual health services increased knowledge about safer sex and relationships and access to contraception[18 38 39]; training staff to recognise, prevent and respond to DRV and sexual harassment led them to act if they saw DRV, promotion of appropriate messages and support led young people to form positive relationships[21 40] and sex and relationships education (SRE) in educational environments increased students' knowledge about risk-taking behaviours, use of sexual health services and contraception.[41 42]

Formative mixed-method multicase research explored the views of FE students, teaching staff, managers and sexual health specialists regarding the acceptability and implementation of the four intervention components within a single complex intervention. The research will inform the development of a new replicable and sustainable intervention to promote safer sex and healthy relationships for young people attending FE settings, which could be rolled out universally.

## METHODS

Data were generated in six FE settings across England (n=3) and Wales (n=3), and one UK sexual health charity, between September and December 2015 using a phased, mixed-method, multicase study design (figure 1). Settings were purposively recruited to reflect different institutional contexts: two 'sixth form' colleges attached to schools (England n=1, Wales n=1), and four large FE college campuses (England n=2, Wales n=2) with yearly intake of >1000 students. All six FE settings invited to take part accepted the invitation. One setting withdrew prior to participation due to practical reasons; this setting was then replaced. The sexual health charity was invited to participate as they are a key provider of services in local communities and educational programmes for children and young people as well as training for professionals and campaigning work across the UK.

### Stage 1: qualitative data generation
Focus groups with FE students (24 groups, n=74 male and n=60 female) and staff (10 groups, n=44 staff), and interviews with FE managers (n=11) and sexual health charity staff (n=12) were used to elicit a broad range of perspectives on if and how the four components should be implemented within a single intervention. Key staff members in each setting identified four single-sex focus groups (two male and two female) of four to eight students aged 16–19 years, and one or two groups of four to eight staff members with varying roles. Qualitative data were transcribed and thematic analysis conducted by two members

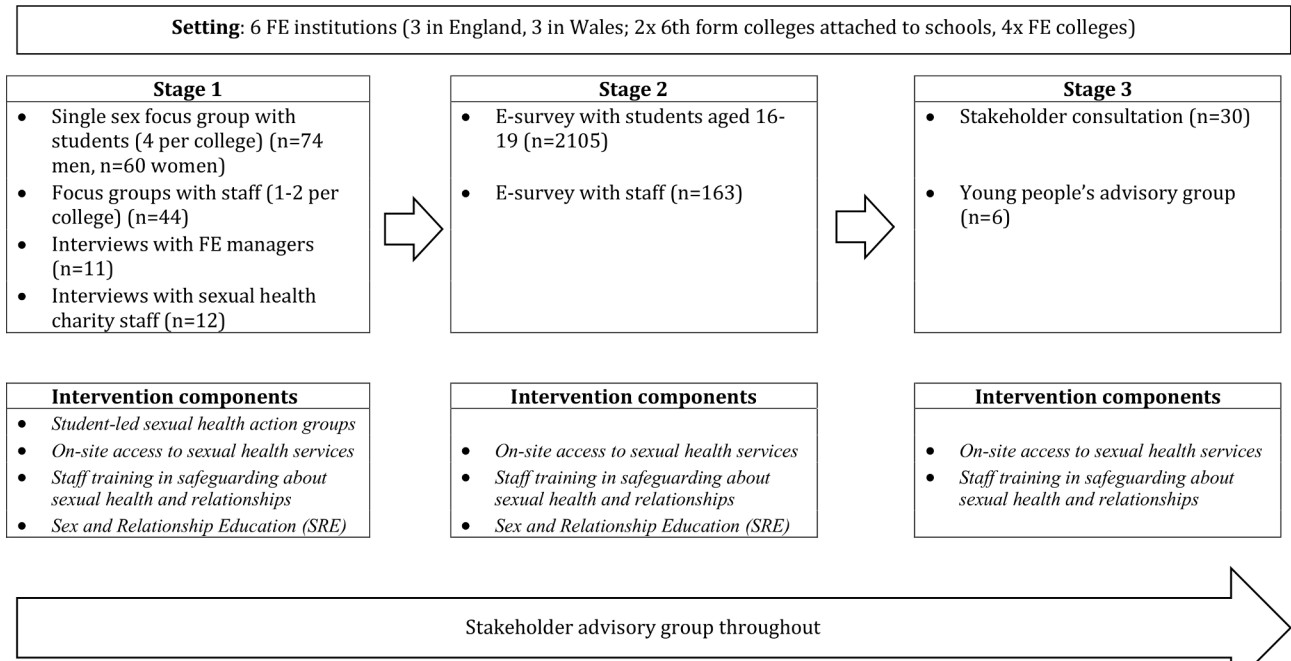

**Setting**: 6 FE institutions (3 in England, 3 in Wales; 2x 6th form colleges attached to schools, 4x FE colleges)

**Stage 1**
- Single sex focus group with students (4 per college) (n=74 men, n=60 women)
- Focus groups with staff (1-2 per college) (n=44)
- Interviews with FE managers (n=11)
- Interviews with sexual health charity staff (n=12)

**Stage 2**
- E-survey with students aged 16-19 (n=2105)
- E-survey with staff (n=163)

**Stage 3**
- Stakeholder consultation (n=30)
- Young people's advisory group (n=6)

**Intervention components**
- *Student-led sexual health action groups*
- *On-site access to sexual health services*
- *Staff training in safeguarding about sexual health and relationships*
- *Sex and Relationship Education (SRE)*

**Intervention components**
- *On-site access to sexual health services*
- *Staff training in safeguarding about sexual health and relationships*
- *Sex and Relationship Education (SRE)*

**Intervention components**
- *On-site access to sexual health services*
- *Staff training in safeguarding about sexual health and relationships*

Stakeholder advisory group throughout

**Figure 1** Mixed-method, multicase design of formative research to inform intervention development.

of the research team (HY and CT). The findings that emerged from the focus groups and interviews were analysed together and identified the candidate components to take forward into stage 2, around which the questionnaires were formed.

## Stage 2: surveys with students and staff

Two self-complete electronic (e)-questionnaires, one with students and one with staff, examined knowledge and use of existing sexual health services, and acceptability of the three components taken forward from stage 1. Data were analysed using STATA. Descriptive statistics are presented in the text as well as the results of chi-square tests to explore gender differences.

### Student survey

Multiple modes of recruitment invited all students aged 16–19 years to participate. Information about the study and a weblink to the e-questionnaire were emailed to all students with an institutional email address. Students also completed questionnaires during scheduled lesson time using electronic tablets, or paper versions of the questionnaire where internet/tablet access was limited, supported by trained fieldworkers. The majority (58%) of questionnaires were completed electronically. The questionnaire measured sociodemographic characteristics and sexual behaviours as well as knowledge, attitudes and experiences of current FE provision and the acceptability of the candidate intervention components identified in stage 1: awareness of and attendance at existing on-site sexual health and relationships services; features of desired on-site services; perceptions of action that FE staff take when safeguarding students in relation to sexual health and relationships; current level of SRE in

FE settings, students' appetite for SRE within FE settings and optimum mode of SRE delivery.

### Staff survey

All teaching and welfare staff (ie, staff employed to deal explicitly with students' health and well-being at FE settings) at each institution were invited to participate in the staff e-survey via a weblink emailed to their institutional accounts. Based on the findings from stage 1, the survey explored staff awareness of current on-site sexual health and relationships services. In relation to safeguarding it explored staff confidence intervening in safeguarding situations, current experience of safeguarding training, appetite for safeguarding training, preferred medium and frequency of delivery, and priorities for staff training. The questionnaire also gathered data on age, gender, teaching experience and role.

## Stage 3: stakeholder consultation

Key findings and recommendations were reported at a stakeholder consultation event with education, health and government professionals and practitioners (n=30), and a young people's advisory group. Breakout discussion groups aimed to finalise the intervention design and explore how to involve stakeholders in the co-production and delivery of an intervention, the content and delivery of safeguarding training, methodological approaches to data collection in FE settings and developing positive FE environments.

### Ethics

Participants were aged 16 years or over and provided consent[43] informed by written descriptions of the study. Students received a £10 voucher for focus group

participation, and student survey participants were entered into a prize draw to win an iPad or £20 voucher.

## Patient and public involvement

A stakeholder advisory group of health and education professionals was established, and consulted at each stage of the research. A young people's advisory group aged 14–25 years was also consulted prior to the funding application to assist in the development of the project aims and research questions. They were also consulted about the content and format of qualitative and quantitative research components, and the final intervention design. A sexual health charity was consulted on the potential components for the intervention, with a representative joining the stakeholder advisory group. The advisory groups reviewed the findings and provided contextual explanation for the results. All stakeholders were invited to the stakeholder consultation event to disseminate the research findings.

## RESULTS
### Stage 1: participant perspectives on four potential interventions components

Student, staff and sexual health charity staff views on the four potential intervention components are summarised below. Participants are coded numerically except for where the identification was uncertain; these are depicted using a '?'.

### Component 1: student-led sexual health action groups
*Student views*

Student reactions to this component were predominantly negative. They reported having neither the time nor inclination to be involved, and did not want to be associated with a group relating to sex and relationships.

> M49: I know it doesn't sound good but in general students can't be asked to do extra stuff like they will literally just do the bare minimum and go home. (FE college 2 (England), male focus group 2)

Several groups of students could see the value of student action groups, but in practice reported a lack of faith that "*anything we say is going to make a difference*" (FE college 2, England, male focus group 1). They discussed difficulty effecting change, based on previous negative experiences of 'student voice' groups.

The majority of students were against the idea of having input into college-level change. However, some did want opportunities to be involved and suggested alternatives such as an anonymous feedback/suggestion box or focus group consultations.

### Staff views

FE staff also felt it was unlikely that students would want to be associated with an action group of this nature, and incentives such as vouchers or letters of thanks were not considered sufficient to engage students.

> S38: I think the problem with students of that age as well is that there's that element of, of embarrassment of kind of, do I really want to be part of this 'cause it's all to do with sex, are people going to think that I'm having sex all the time?' (FE college 1 (Wales), staff focus group 1)

Staff argued that the transient nature of FE was a barrier to student engagement in such activities. Buy-in from the wider staff body was considered important for the successful implementation of student voice campaigns that were not '*set up to fail*' (FE college 1 (England), staff focus group 1). However, they felt that due to reductions in funding, FE settings rarely had sufficient infrastructure or resources to support these.

### Component 2: on-site sexual health and relationship services
*Student views*

Students responded positively to the idea of sexual health and relationship services on-site, reporting that these could address many barriers preventing young people engaging with typical service provision.

> F55: It can be quite annoying, if you can't drive, to get there and have (the contraceptive implant). It took me ages to get an implant, so if there was easy access, I think loads of people would have them. 'Cause it's definitely hard to go by yourself, if you don't want your parents to know … so I think a lot of people would be more safe if it was easier to get it'. (FE college 1 (Wales), female focus group 1)

The services students wanted commonly included free access to a range of contraception, STI screening/testing, pregnancy testing and emergency contraception.

Negative social norms relating to communicating about sex and sexual health were a significant barrier to accessing sexual health services by the majority of students. This was discussed in relation to: embarrassment interacting with sexual health service staff, particularly outside the service; fear of being seen accessing services by other students and concerns about service staff maintaining confidentiality. The location of services was crucial to avoid embarrassment and encourage attendance. Students acknowledged the risk that a private location could perpetuate the view that using sexual health services is taboo or shameful, so wanted services to be accessible but discreet.

> M19: Just basically so you're not embarrassed to go up because other students might see you and they might laugh at you and whatever.
>
> Facilitator: How do you think that would work?
>
> M18: Yeah (M19) mentioned about just making sure it's away from other students.
>
> M19: Having a secluded place in the school.
>
> M16: It's got to be anonymous, there's still a fear that the school might phone your parents or something, just say if you're coming for condoms and you don't

want your parents finding out or your friends finding out. (Sixth form (England), male focus group 2)

To encourage attendance, students highlighted the importance of sexual health service staff being knowledgeable, trustworthy, non-judgemental and easy to relate to. Students favoured services run by the same staff with whom they could build a relationship, but not have regular interactions with outside the service. The vast majority of students did not think teachers should be involved due to bad relationships with teachers, embarrassment, lack of trust and actual or perceived lack of relevant knowledge.

F2: I wouldn't want to talk to a teacher about any of like my problems.

F5: Like if they've seen you around college or something, that would just be horrible.

F1: I'd want like a qualified person.

F3: Somebody who's got their head on, like you know what they mean and they understand.

F5: But it's also good that they're talking about just like general…

F3: Someone who's down to earth.

All: Yeah.

F1: Someone enthusiastic but caring. And not judgmental either.

F2: Like if that's job like a welfare officer's, it's a bit different to a teacher, that's a sort of 'how the head works', you know what I mean? Like everyday sort of life thing, not just teaching Maths, English, whatever. So someone a bit more comfortable. (FE college 1 (England), female focus group 2)

The frequency and timing of the service were particularly relevant for students with busy timetables, few hours in college or intermittent attendance. They saw value in a drop-in service that ran several times each week, at different times of day. More frequent services were also believed to help them feel less self-conscious about attending.

F27: Open more than once a week, ours is only open on the Tuesday or something?

F28: Yeah, you literally know why they're there.

F24: Both, yeah, because you're not going to want to queue up knowing that everyone's there for the same reason. (Sixth form (England), female focus group 2)

Publicising the service was important to maximise student awareness of on-site services. Students' suggestions for communication included college induction; email; text; social media; via tutors or registration time; college websites, computers or noticeboards; leaflets and posters in public areas and toilets around the setting. Students also highlighted the potential of incorporating digital communication such as advice and information online.

### Staff views

Staff and charity workers wanted students to have '*a safe place to go and talk to someone*' (FE college 1 (England), staff focus group 2). Staff suggested that on-site services could provide a unique opportunity for students facing barriers such as transport, embarrassment and not wanting parents to know about their sexual activity.

Providing a range of sexual health and support services was reiterated by staff, including managers, as well as sexual health charity staff.

C3: In a perfect setting, we'd want to see every school and college has the facility to prescribe contraception, chlamydia tests, give condoms out, pregnancy test at the minimum, as a minimum. (Sexual health charity staff (interview 3))

It was argued that to provide meaningful, preventative care, on-site services needed staff to offer sexual health and relationships advice, information (eg, on sexual health, emotional/psychosocial side of sex, mental health, violence/abuse) and counselling.

S26: I think it's all well and good to give out free condoms…but then are you really dealing with the issue…that doesn't cover things like rape or, when it, when no means no, for some people… there needs to be a level of, you know, 'why' behind it…why they feel the need to go out and have as many relationships as possible, or why they feel the need to put those sexy photographs on Tinder and go off with people they don't know…it's those questions that really need…that they need to answer for themselves with guidance and help in order to ultimately long term protect themselves. (FE college 2 (Wales), staff focus group 2)

Similar methods of publicising on-site services were supported by staff and charity workers, with the addition of repeating publicity information. Staff were willing to be a medium for delivery, but wanted support ensuring messages were clear and consistent. They supported the idea of incorporating digital communication into both on-site services and education.

Staff and charity workers concerns related to the sustainability and funding of on-site services, as well as support from FE staff.

M7: I think and I know that I won't be able to have it here but I really would like to have a counsellor on-site….Ideally, we should have a nurse on-site who could, perhaps, have a dual role. That would be fantastic if we could, but there's no money in the pot for that, at this moment in time. (FE college 2 (Wales), manager interview)

C6: You also need to ensure that you have the full support of the college, sixth form, whatever because without that you're not going to get anywhere at all. (Sexual health charity staff (interview 6))

### Component 3: staff training in safeguarding specifically relating to safe sex and healthy relationships
*Student views*

Students discussed a small number of instances where staff effectively addressed safeguarding concerns relating to sex or relationships. However, more commonly, students reported a college environment where staff did not take appropriate action over instances of sexual harassment, sexualised name-calling and other sexist behaviour.

> F?: The harassment with the whistling.
>
> F?: There's always tutors around though, because it happens at lunch and stuff, and they never say anything, and they don't just whistle, they like make comments as well. (FE college 2 (Wales 2), female focus group 1)
>
> M35: Our tutor takes the piss out of us. (laughter).
>
> M39: Yeah like, say you come into college and say I shagged someone in the shed, he'll take the piss out of you straight away.
>
> M40: He'd be like, high five.
>
> M35: He'd just like laughing. (FE college 2 (England), male focus group 1)

When discussing how such instances should be managed, the majority of students said that staff needed to be able to identify and respond appropriately if a student was at risk of harm. Both young men and women acknowledged that sexual harassment and sexualised name-calling happened, but they did not always identify this as problematic. Students reported that staff needed support distinguishing harmful behaviour from 'banter'.

> F?: I think they should learn about like dealing with disrespect, because boys are like that. But I think they should deal with it more than they do now.
>
> F?: And like saying the right thing to them as well, like, if you're like some teachers could just say, oh they're just messing, don't worry about it but other people could take it out, so like maybe training on like what to say, like how to deal with it. (FE college 1 (Wales), female focus group 2)
>
> M43: When you're in their lessons and they see something wrong they could report it or whatever, tell someone and then give you advice or something.
>
> M44: Keep you back after to speak to you about it.
>
> M49: I don't know like if they do something abusive then they should be able to pick up on it earlier.
>
> M?: See the signs and stuff. (FE college 2 (England), male focus group 2)

Students felt that all staff should be trained in this area. They wanted to be able to raise concerns with any staff member, and know that this would be dealt with appropriately, ideally by the same member of staff.

> F?: If you're actually going to a member of staff with something, and um, you feel comfortable talking to

them, then they could sometimes be like, well, this is not for me, this is for someone else.

> F?: Being able to talk to someone you want to talk to and for them to say, like, you know, for them to be able to deal with it, instead of going to someone else.
>
> F?: Yeah. It's like our tutor says she's not qualified to deal with bullying, so we'd have to go to somebody else. She has actually said that. (FE college 1 (Wales), female focus group 1)

*Staff views*

Staff reported that they needed additional training responding to student disclosures, and identifying and addressing safeguarding concerns relating to sex and relationships. Staff were keen to undertake more preventative action, rather than current safeguarding which often related to crisis situations.

> S?: I always think it's such a big thing for a 16, 17 year old to say 'Actually, can I have help with that', or like 'Can I just ask someone about that'…
>
> S?: So anyone who gets asked should just be able to deal with it but…
>
> S?: Should be able to say, 'Yep, right I know exactly what you need to do' (FE college 1 (England), staff focus group 2)

Staff also felt that all colleagues should be trained in this area. However, they acknowledged that some would resist additional training or responsibilities, as sex and relationships would be *'one area where they'll say "I didn't sign up for this"'* (FE college 1 (England), staff focus group 1). Both FE and charity staff felt that staff support was crucial to the success of training programmes, and suggested linking safeguarding with formal inspections to encourage uptake.

### Component 4: sex and relationships education
*Student views*

Students identified 'gaps' in the SRE they had experienced in secondary school, including how to recognise and deal with STIs; pregnancy and abortion; violence and abuse (including emotional abuse); consent; emotional aspects of sex; pornography; digital safety; sexual pleasure; revenge porn; social media and communication and diverse gender, sexual identities and relationships. However, the key message from students was that FE was too late to address these gaps; SRE needed to be delivered in secondary school, if not earlier.

Students also noted the varied knowledge and experience that they brought entering FE, which they felt would be a barrier to engagement in SRE lessons. Students reported that they would not attend SRE lessons, even if compulsory, due to perceived ineffectiveness, other competing demands and lack of space in their timetable.

> F50: I don't think it would work perhaps as a lesson … I don't think it's a good idea.

F49: It would feel like you're being forced to do something you don't want. Have you heard of PHSE? No one ever listens and the parables are crap … and the teachers don't know what they're doing.

F50: It's just an hour to sit there and talk to your mates.

F49: And you actually take the topic less seriously afterwards. So I think it had a negative effect. (FE college 2 (England), female focus group 2)

M2: Wouldn't go.

Facilitator: You wouldn't go. Even if it was compulsory.

M2: Wouldn't go. (FE college 2 (Wales), male focus group 1)

Finally, students perceived their FE tutors to be lacking in the training, knowledge and credibility needed to deliver effective SRE.

### Staff views
Staff and charity workers agreed that SRE delivery in FE was too late for students' sexual health and relationships needs, and shared student views that most teachers were ill equipped to deliver effective SRE.

S37: Just because we're adults and professionals, doesn't mean we know about healthy relationships … we'd need some kind of external training.

S42: You see that's the difficulty of the thing you see when you've got tutors doing it and they are that close to the student I just don't think it fits right. (FE college 2 (England), staff focus group 1)

Staff also acknowledged the wide disparity in students' knowledge and skills entering FE. Additionally, they noted the risk of student disengagement by repeating topics covered in school; and logistical challenges of organising SRE lessons in FE settings, given students' differing contact hours and timetables.

### Stage 2: surveys with students and staff
A total of 2105 students aged 16–19 years participated. The majority (54%, n=1137) were female, heterosexual (87%, n=1829) and on a non-academic pathway (59%, n=1245). Under a fifth of the sample were from black and minority ethnic groups (14%, n=292) and around a third had <£20 to spend on themselves each week (35%, n=742).

A total of 163 staff responded with an even distribution across ages 20–60 years. The majority were female (72%, n=115) and had worked at that FE setting for >5 years (72%, n=106). Over a third of respondents were subject teachers (38%, n=56), under a fifth were welfare staff (15%, n=25) and around a quarter had other roles (eg, administration staff, learning support and technicians) (28%, n=45).

### Component 1: student-led sexual health action groups
Low enthusiasm from students and staff meant that this component was not taken forward to stage 2.

### Component 2: on-site sexual health and relationship services
*Students*
All FE settings had some form of sexual health and relationships service, yet student awareness of service provision was poor: 77% (n=1299) of students did not know if their college provided STI testing, 68% (n=1330) pregnancy testing, 66% (n=1294) contraception, 47% (n=935) condoms and 46% (n=911) advice.

Use of services was also low: 91% (n=1789) of students (88% of sexually active students, n=1135) had never attended their on-site services. However, 49% (n=870) (44% of sexually active students, n=580) reported that they would attend an on-site service if it were freely available and not run by teachers.

When students were asked what services they wanted at their college, nearly two-thirds wanted condoms (64%, n=1205) and advice, support, information and/or counselling (63%, n=1180), followed by emergency contraception (47%, n=894), pregnancy testing (46%, n=868), other contraceptives (46%, n=859) and STI testing (46%, n=866). The largest proportion of students wanted services available after college (48%, n=862) followed by lunchtime (41%, n=731).

*Staff*
Over a third of staff (35%, n=43) did not know if services were available for their students.

### Component 3: staff training in safeguarding specifically relating to safe sex and healthy relationships
*Students*
Less than half of students (44%, n=807) agreed that staff took appropriate action to stop students calling each other offensive names, such as slut or slag. Significant gender differences were found; 51% of men and only 37% of women agreed that staff took appropriate action ($X^2$=32.056, p<0.001). When students were asked if they would speak to a member of staff about DRV if it was happening to someone within or outside college, 38% (n=707) and 36% (n=671) agreed, respectively.

*Staff*
The majority of staff reported feeling confident intervening if they saw a student: being called sexually offensive names (90%, n=116); being unwantedly touched, groped or kissed (87%, n=112); with a sexually explicit image of another student on their phone (83%, n=107) or watching pornography on their mobile phone, tablet or laptop (83%, n=107).

However, less than half of staff received training in safeguarding specifically about sexual health and relationships (47%, n=55). The majority reported that all staff should be trained (67%, n=84), that training should be compulsory (75%, n=93), happen once a year (35%, n=57) and be delivered face-to-face (47%, n=76).

When asked for their training priorities, around 90% wanted training on identifying safeguarding concerns in DRV (n=112); responding appropriately to DRV (n=111); preventing sexualised language/behaviour at FE settings (n=111); sending sexually explicit images ('sexting'). Around 80% wanted training on young people's use of pornography (n=107); answering questions about sexual health (n=102) and consent, sex and the law (n=103).

### Component 4: sex and relationships education
*Students*
Around one in five students (21%, n=416) reported that their FE setting taught them about safe sex, and 20% (n=381) about healthy relationships. A total of 33% (n=629) reported that their FE setting taught them what to do if students call other students sexually offensive names, and 54% of students wanted to be taught about this topic. Over a quarter of students (28%, n=538) reported that FE taught them about safety when online dating and 54% (n=1004) wanted teaching on this. Over a third of students (35%, n=649) reported learning about sexual consent in FE, whereas 57% (n=1047) wanted teaching on this. Similarly, almost a third of students (32%, n=601) reported that FE taught them who to go to if they or a friend experience dating or relationship violence, whereas 60% of students wanted to be taught on this topic.

### Stage 3: synthesis of results and consultation with key stakeholders
Following stage 1, student-led sexual health action groups were not considered appropriate, sustainable methods of intervention for FE settings. The same conclusion was drawn about SRE following stage 2. On-site sexual health and relationship services and staff training in safeguarding about sexual health and relationships were important, appropriate gaps in FE provision and welcomed by staff, students and health professionals. These findings were summarised and presented at a stakeholder consultation event, and with the young people's advisory group. Breakout sessions discussed recommendations for the optimum delivery and implementation of the intervention (table 1). Feedback was incorporated to finalise the intervention design.

### DISCUSSION
Of the four components examined, on-site sexual health and relationships services and staff training in safeguarding about sexual health and relationships were acceptable, appropriate gaps in current FE provision that could be implemented on a sustainable basis and would be welcomed by staff, students and health professionals. On-site services were available in most FE settings but few students or staff were aware of these, and around 90% of sexually active students had never visited an on-site service. However, almost half reported that they would attend. Concerns about staff support for the delivery of services were not unique to FE, nor to an intervention of this nature.[44] Almost all staff reported feeling confident intervening with safeguarding issues relating to sexual health and relationships, but only 44% of students agreed that staff took appropriate action. This may reflect differences between staff and students on whether harassment and sexualised name-calling was problematic, as well as a lack of staff training about how to intervene. Less than half of staff reported having received safeguarding training about sexual health and relationships but most wanted compulsory training for all staff.

Student-led sexual health action groups and SRE were not considered acceptable interventions for FE settings. Contrary to school-based literature using student-led action groups[36 37] but consistent with other FE-based health interventions,[44] students lacked motivation to engage in student-led action groups, felt there were more appropriate ways for their 'voice' be heard and that this topic would be more suitable for less transient student settings. This component was therefore not taken forward after stage 1. Although 50%–60% of students wanted to be taught about issues relating to sexual health and relationships in FE, the setting was overwhelmingly considered 'too late' for SRE delivery, and too challenging given the diversity of FE settings and students' varied sexual health knowledge, skills and experience. This is consistent with existing literature highlighting the varying quality and quantity of SRE in schools which young people believe is currently delivered too late.[45] This component was not taken forward after stage 2 (see Table 1

The study is not without its limitations. FE settings were not always able to provide the numbers of enrolled students, and when provided, the numbers do not reflect attendees on-site. This prevents the calculation of an accurate response rate or sampling frame. Collecting a random or representative sample of students or staff in FE settings posed significant challenges due to students' varied patterns of attendance. Selection bias may have operated, such that students who have strong opinions on sexual health and relationships may have been more likely to respond, potentially resulting in findings that are unrepresentative of the wider population. Some of the results may be setting specific, the findings therefore warrant further optimisation, refinement and piloting before wider implementation. Similarly, the work was conducted in the UK, therefore the provision of sexual health services may differ to other international contexts.

The development and evaluation of comprehensive sexual health and relationship interventions is recognised as a public health priority[13–20]; however, little attention is paid to the developmental phase of the complex intervention cycle[35] and to date, FE-based sexual health and relationship intervention activities have been ad hoc. This formative multicase study addresses gaps in substantive and methodological knowledge, providing information on the acceptability of intervention components, theorising the mechanisms of change and how implementation and causal pathways may vary by context. This study

**Table 1** Summary of results from stages 1, 2 and 3.

| Component | Stage 1: interviews/focus groups | | Stage 2: e-survey | | Stage 3: stakeholder consultation |
|---|---|---|---|---|---|
| | Students (24 focus groups, n=74male and n=60female). | Staff (10 further education (FE) staff focus groups, n=44, 11 FE manager interviews and 12 sexual health charity staff interviews). | Students (n=2105). | Staff (n=168). | Educators, health and government professionals and practitioners (n=30) and a young people's advisory group. |
| Student-led sexual health action groups | Largely negative. Students are too busy. Student do not expect groups to be effective. Students do not want to take part in groups of this nature. | Staff perceive multiple barriers including: student embarrassment, engagement, motivation and cohort transience. Incentives not sufficient. Limited college funding and support. | Low enthusiasm from student and staff meant that this component was not taken forward to stage 2 and was therefore not discussed at stage 3. | | |
| On-site access to sexual health services | Largely positive. Desired provision: free contraception, STI screening, pregnancy testing and advice. Accessible but discreet location. Knowledgeable, trustworthy, non-judgemental, consistent staff who students can relate to. Drop in service several times a week at varied times of the day. Well publicised via college staff, digital and social media. | Largely positive. Offering a range of contraception and testing services, and inclusive of advice, support and emotional care. Support to publicise services. Sustainability for on-site services (financial and staff support). | % students did not know if their college provided: 77% STI testing; 68% pregnancy testing; 66% contraception; 47% condoms; 46% advice; 88% of sexually active students had never attended on-site services; 44% of sexually active students would attend an on-site service if freely available and not run by teachers. % of students wanting services: 63% advice, support, information or counselling; 64% condoms; 47% emergency contraception; 46% pregnancy testing; 46% other contraceptives; 48% wanted services after college; 41% wanted services during lunchtime. | 35% of staff did not know if sexual health and relationships services were available for their students. | Deliver a range of contraceptive, testing and advice and support services by a trained youth friendly, health professional in a way that is non-stigmatising and promotes confidentiality. The services need to be open at least twice a week and located in an accessible but anonymous location. Services need to be well publicised to increase student and staff awareness. Digital messages with information and signposting should be incorporated into publicity. |

Continued

**Table 1** Continued

| Component | Stage 1: interviews/focus groups | | Stage 2: e-survey | | Stage 3: stakeholder consultation |
|---|---|---|---|---|---|
| | Students (24 focus groups, n=74male and n=60female). | Staff (10 further education (FE) staff focus groups, n=44, 11 FE manager interviews and 12 sexual health charity staff interviews). | Students (n=2105). | Staff (n=168). | Educators, health and government professionals and practitioners (n=30) and a young people's advisory group. |
| Staff training in safeguarding about sexual health and relationships | Largely positive. Students wanted staff to be able to identify and respond appropriately. Students wanted staff to be able to distinguish 'banter'. Students wanted all staff to be trained so that they can approach any member. | Staff need support and training when responding to safeguarding issues on these topics. Staff want to take preventative action. All colleagues to be trained. Raised concerns about staff engagement in training and implementation of safeguarding. | 44% agreed that staff took appropriate action to stop students calling each other offensive names such as slut or slag. 38% would speak to a member of staff about dating or relationship violence if it was happening to someone in college. 36% would speak to a member of staff about dating or relationship violence if it was happening to someone outside college. | % staff who felt confident intervening if they saw: 90% a student being called offensive names; 87% being unwantedly touched, groped or kissed; 83% with a sexually explicit image of another student on their phone; 83% watching pornography on their mobile phone, tablet or laptop; 47% received safeguarding training specifically about sexual health and relationships; 67% wanted all staff to be trained in safeguarding about sex and relationships; 75% wanted compulsory staff training; 35% wanted training yearly; 47% wanted face-to-face training. Staff training priorities: 90% identifying safeguarding training in DRV; 90% responding appropriately to DRV; 89% preventing sexualised language/behaviour at FE settings; 86% sending sexually explicit images; 86% young people's use of pornography; 83% answering questions about sexual health; 83% consent, sex and the law. | Staff training needs to be delivered to all members of FE staff. It would need to be face-to-face and cover topics such as recognising signs of dating and relationship, and gender-based violence, how to take appropriate action when faced with students presenting with these issues, and how to signpost students to appropriate services. |

Continued

**Table 1** Continued

| Component | Stage 1: interviews/focus groups | | Stage 2: e-survey | | Stage 3: stakeholder consultation |
|---|---|---|---|---|---|
| | **Students (24 focus groups, n=74male and n=60female).** | **Staff (10 further education (FE) staff focus groups, n=44, 11 FE manager interviews and 12 sexual health charity staff interviews).** | **Students (n=2105).** | **Staff (n=168).** | **Educators, health and government professionals and practitioners (n=30) and a young people's advisory group.** |
| Sex and Relationships Education (SRE) | SRE in FE was overwhelmingly considered too late in young people's lives. Students wanted a wider range of SRE, not just focussing on STIs and contraception. SRE delivery by knowledgeable, non-judgemental, easy to relate to staff. Students anticipated lack of engagement in SRE lessons. | Students should receive SRE earlier in their education. Staff felt the lacked knowledge, training and credibility to deliver SRE to students. Barriers to SRE delivery included varied student knowledge and experience, student engagement and timetabling issues. | % who felt their FE setting taught them about: 21% safe sex; 20% healthy relationships; 33% what to do if students call other students sexually offensive names; 28% safety when online dating; 35% giving consent when having sex; 32% who to go to if they or a friend experience forms of DRV. % who wanted their FE setting to teach them about: 54% what to do if students call other students sexually offensive names; 54% safety when online dating; 57% giving consent when having sex; 60% who to go to if they or a friend experience forms of DRV; 82% felt that SRE should be taught by specialised SRE/health education staff; 61% felt SRE should be taught by external organisations. | | SRE delivered at FE level was generally considered too late for young people and was therefore not discussed at stage 3. |

design is applicable to other outcomes and settings where the acceptability, priority to stakeholders and barriers to sustained implementation of multiple candidate intervention components is unknown. Future research is required to explore if and how the remaining components can address health inequalities, as hypothesised in the logic model (online supplementary appendix 1).

In conclusion, an intervention comprising on-site sexual health and relationship services and staff training in safeguarding about sexual health and relationship was perceived to address important gaps in current FE provision, and to be acceptable and wanted by staff, students and sexual health professionals. These components should be combined in a universal, multilevel intervention to improve safe sex and healthy relationships in FE settings.

**Acknowledgements** RL has been supported by the UK Medical Research Council (grant MC_UU_12017/11), and Chief Scientist Office (grant SPHSU11). The authors would like to thank the sexual health charity, the FE college staff and students for their time and contribution to the project, and members of the stakeholder advisory group for their guidance throughout the project. The authors would also like to thank Professor Adam Fletcher for his work on the project.

**Contributors** HY, CT and JW conceptualised the paper. HY and CT led the analysis and interpretation of the data with support from JW, RL and CB. HY wrote a first draft of the manuscript. HY, JW, CB, RL and CT contributed to subsequent revisions of the paper and approved the final manuscript.

**Funding** The SaFE Project is a partnership between DECIPHer at Cardiff University, The London School of Hygiene and Tropical Medicine, The Institute of Education and a sexual health charity, funded by the Medical Research Council Public Health Intervention Development Scheme (MRC PHIND) (grant number MR/M026272/1, since October 2017). The work was undertaken with the support of The Centre for the Development and Evaluation of Complex Interventions for Public Health Improvement (DECIPHer), a UKCRC Public Health Research Centre of Excellence. Joint funding (MR/KO232331/1) was from the British Heart Foundation, Cancer Research UK, Economic and Social Research Council, Medical Research Council, the Welsh Government and the Wellcome Trust, under the auspices of the UK Clinical Research Collaboration.

**Competing interests** None declared.

**Patient consent for publication** Not required.

**Ethics approval** Ethical approval was provided by Cardiff University School of Social Sciences Research Ethics Committee.

**Provenance and peer review** Not commissioned; externally peer reviewed.

**Data sharing statement** Data are available on request.

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
