## [Reviewer comments · BMJ Open]

ARTICLE DETAILS

TITLE (PROVISIONAL)	Formative mixed method multi-case study research to inform development of a safe sex and healthy relationships intervention in Further Education (FE) settings: The SaFE Project
AUTHORS	Young, Honor; Turney, Catherine; White, James; Lewis, Ruth; Bonell, Christopher

VERSION 1 - REVIEW

REVIEWER	Jennifer Tsai University of Southern California, United States
REVIEW RETURNED	23-Oct-2018

GENERAL COMMENTS	This study examines current gaps in current sexual health curriculum and proposes ways to improve sexual health effectiveness. Overall, the manuscript is well written and easy to follow. However, the methods section, in particular, needs additional information; there are also a few grammatical errors that should be addressed throughout the manuscript. Specific comments are attached. BMJ Open: Formative Mixed Method Multi-case Study Research to Inform Development of a Safe Sex and Healthy Relationships Intervention in Further Education (FE) Settings: The SaFE Project This study examines current gaps in current sexual health curriculum and proposes ways to improve sexual health effectiveness. Overall, the manuscript is well written and easy to follow. However, the methods section, in particular, needs additional information; there are also a few grammatical errors that should be addressed throughout the manuscript. Introduction: - Pg. 3, Line 24: The authors mention that multi-component interventions are less likely to generate inequalities. Additional explanation regarding how multi-component interventions can reduce inequalities (and which inequalities) can help with the justification of the study.- Pg. 3, Line 26: The authors mention that FE settings are the best places to implement these types of programs. For someone unfamiliar with the UK system, it would be helpful to have a short definition of FE (how is this different from normal high schools?) and what the characteristics of these students are compared to other educational institutions (e.g., high-risk adolescents for sexual behaviors?).
---

- The inclusion of the project name, SaFE Project, should be mentioned somewhere in the introduction.

Methods:

- Pg. 4, Line 17-21: There is no mention of the sexual health charity setting where data also seems to have been collected. In addition, a stronger justification for why a sexual health charity was approached as part of this study should be included.

- The method for implementing and analyzing data for student e-surveys is unclear. Were these a series of yes/no, fill-in questions, or did a facilitator conduct online interviews for open-ended questions?

- Pg. 4, Line 33: It says Descriptive statistics are presented and chi-square tests explored gender differences, but I don't see a table of descriptive statistics (other than information regarding gender) and there is no discussion of the chi-square results. These should be included and discussed in the manuscript.

- Methods for how the quantitative and qualitative analyses were conducted should be detailed in the Methods section for each group of participants identified (i.e, student survey, staff survey, focus group). For the qualitative analyses, it would also be helpful to know how themes were identified, who reviewed qualitative responses, and if themes were triangulated with other sources (e.g., observations during focus groups, etc).

Results:

- Pg 6, Line 46-47: The authors identified these quotes as coming from "School". Does this mean Charity or is this another FE school? What is the difference between "School Focus Group" and "FE Focus Group"? Explanation of this somewhere in the text would help the readers where this information was collected.

- Pg. 8, Line 47, 50: Throughout the results section, some of the quotes have F? or M? Not sure what these "?" mean. In addition, some have numbers attached and some do not. These should be consistently presented.

- Pg. 9, Line 27-31: Similar to the previous comment. Why use S? Since we know that these quotes are from Staff focus group, perhaps it would be best to present these as M/F as well.

Discussion:

- A general discussion regarding the feasibility of integrating these sexual health components to curriculum should be discussed and how these may be incorporated into current sexual health curriculum can provide more informative next steps for sexual health programs.

- The inclusion of a few points about how these components address health inequalities mentioned in the introduction can further justify the impact of this work.

REVIEWER	Christine Markham UTHealth School of Public Health, USA
REVIEW RETURNED	21-Nov-2018

GENERAL COMMENTS	The authors present original findings from a mixed method multi-case formative research study to design and assess the feasibility of a safe sex/healthy relationships intervention for further education settings in the UK. Although the study comprises a small number of sites (n = 6), it provides potentially generalizable findings to guide further intervention testing. Overall, the study appears to have been well-conducted, and the manuscript is well-written; however, some findings would benefit from clarification, and some terms would benefit from translation for a non-UK readership. The following comments are offered to strengthen the manuscript. Methods, p.4. Address any issues in site recruitment. Only 6 sites participated in the formative research study. How many sites were approached for recruitment but declined to participate due to the potentially controversial (sexual) nature of the study? This seems to be a major question that is not addressed in the study. Overall, what is the acceptability among further education (FE) settings to address sexual health among their student population? Methods. P.4. For non-UK readers, clarify what is a 'sexual health charity'? Is this something similar to, say, Planned Parenthood in the US? That is, a non-profit organization that provides sexual health education to other youth settings? Methods. P.4. Similarly, what does 'staff safeguarding action' mean outside of the UK? This needs translation for non-UK readers. Also, 'welfare staff' – does that translate to counseling staff? Results, Stage 2, page 8. It seems that multiple staff brought up issues of sustainability (cost, available funding) for on-site sexual health care services, and for administrative support in FE settings. How widely voiced were these concerns across staff focus groups? In Stages 2 & 3, it seems that these staff concerns were overlooked or not addressed again, when they seem as if they could be major feasibility barriers to the implementation of on-site sexual health and relationship services. Results, Stage 2. Clarify, what was the response rate for student surveys, and for staff surveys. Again, it would be good to clarify in Stage 1, what was the overall response rate for invited sites that agreed to participate, to allow readers to assess the generalizability of these results across typical FE settings. Results, Stage 3. Clarify why Sex and relationships education (SRE) was dropped at this stage, when findings from the Stage 2 student surveys indicated that ~50-60% of students would use these types of services. Conversely, clarify why 'on-site sexual health and relationship services' were considered feasible and sustainable for further consideration, when staff input in Stage 1 questioned their sustainability and administrative support. Of all the potential intervention strategies proposed, this category seems the most unsustainable and potentially politically fraught, at least in non-UK settings. Clarify why the advisory group gave this strategy the "green light". What economic and political resources would need to be in
--

	place across the UK to make this a viable strategy? How feasible are these resources to secure across the board? Discussion. Clarify why strategy 4 Sex and relationships education was demoted in final deliberations, when it seemed to rate fairly high in Stage 2 student surveys (~50-60+% of students endorsed specific topics). Clarify, what weight did discussion with the adult and youth advisory boards have over Stage 2 constituent survey findings? Why so? Figure 1. Include response rates for student and staff survey participation. Table 1, Stage 3. Did the stakeholder consultation address resource/administrative support issues for On-site access to sexual health services that were raised in the Stage 1 staff focus groups & interviews? This seems to be a potentially major oversight that is not addressed in the narrative. Table 1, Stage 3, sex and relationship education. What evidence is provided from stakeholders to support the premise that SRE delivered at the FE level is “too late” given that the majority of student survey respondents in Stage 2 endorsed this strategy?
--	--

REVIEWER	Adela Montero Associate professor Center for Reproductive Medicine and Integral Development of Adolescents Faculty of Medicine University of Chile
REVIEW RETURNED	02-Feb-2019

GENERAL COMMENTS	Congratulations for this research that addresses very important issues related to the sexual and reproductive health of adolescents and young people. I would like to comment that several of their findings have also been evidenced in my country, for example, the existing taboo regarding sexuality. In Chile, confidentiality to access sexual health services is very important for adolescents and young people as well as the need to have trained, empathic professionals, committed to the well-being of adolescents and young people. Thank you very much for this great work.
---

VERSION 1 – AUTHOR RESPONSE

Reviewer: 1

Reviewer Name: Jennifer Tsai

Institution and Country: University of Southern California, United States

Please state any competing interests or state 'None declared': None declared.

Please leave your comments for the authors below. This study examines current gaps in current sexual health curriculum and proposes ways to improve sexual health effectiveness. Overall, the manuscript is well written and easy to follow. However, the methods section, in particular, needs

additional information; there are also a few grammatical errors that should be addressed throughout the manuscript.

Introduction:

- Pg. 3, Line 24: The authors mention that multi-component interventions are less likely to generate inequalities. Additional explanation regarding how multi-component interventions can reduce inequalities (and which inequalities) can help with the justification of the study.

Apologies we are not clear about which inequalities this reference refers to. To clarify, this citation refers to evidence that multi-component interventions do not exacerbate socio-economic inequalities. There is limited evidence that multi-component interventions can reduce inequalities. The text now reads "Cochrane and Campbell reviews^{24,25} and NICE guidance²⁶ recommend further research on multi-component interventions which tend to be more cost-effective²⁷ and are less likely to generate socio-economic inequalities.²⁸"

- Pg. 3, Line 26: The authors mention that FE settings are the best places to implement these types of programs. For someone unfamiliar with the UK system, it would be helpful to have a short definition of FE (how is this different from normal high schools?) and what the characteristics of these students are compared to other educational institutions (e.g., high-risk adolescents for sexual behaviours?).

Thank you for this comment. We have clarified the context of FE, including the characteristics of the students, as well as giving comparable settings in the US or Australia. The text now reads "Further education settings, akin to technical and further education in Australia and community colleges in the United States, primarily serve 16-19-year-olds. Socio-economically diverse and of a broader age range than in university settings, they provide an optimal setting for such work."

- The inclusion of the project name, SaFE Project, should be mentioned somewhere in the introduction.

We have now included the name of the project in the introduction. The text now reads "The SaFE Project aimed to identify intervention components to promote safe sex and healthy relationships in FE settings which were acceptable, perceived to be a priority for students and FE staff, and could be implemented sustainably."

Methods:

- Pg. 4, Line 17-21: There is no mention of the sexual health charity setting where data also seems to have been collected. In addition, a stronger justification for why a sexual health charity was approached as part of this study should be included.

Thank you for identifying this omission. We have now clarified in the text to mention that we collected data from the sexual health charity during the time period of the project, and to justify why we approached the charity as part of the study. The text now reads "Data were generated in six FE settings across England (n=3) and Wales (n=3), and one UK sexual health charity, between September and December 2015 using a phased, mixed-method, multi-case study design (Figure 1). Settings were purposively recruited to reflect different institutional contexts: two 'sixth form' colleges attached to schools (England n=1, Wales n=1), and four large FE college campuses (England n=2, Wales n=2) with yearly intake of >1000 students. The sexual health charity was invited to participate as they are a key provider of services in local communities and educational programmes for children and young people as well as training for professionals and campaigning work across the UK."

- The method for implementing and analyzing data for student e-surveys is unclear. Were these a series of yes/no, fill-in questions, or did a facilitator conduct online interviews for open-ended questions?

We have clarified the data collection methods; these were self-complete electronic (e)-questionnaires. We have also clarified how data were analysed; data were analysed using STATA to summarise responses. The text now reads “Two self-complete electronic (e)-questionnaire, one with students and one with staff examined knowledge and use of existing sexual health services, and acceptability of the three components taken forward from stage 1. Data were analysed using STATA.”

- Pg. 4, Line 33: It says Descriptive statistics are presented and chi-square tests explored gender differences, but I don't see a table of descriptive statistics (other than information regarding gender) and there is no discussion of the chi-square results. These should be included and discussed in the manuscript.

We would like to clarify that the descriptive statistics and chi-square tests are presented in the text. For example, page 11 details the chi2 analysis. The presentation of these findings in the text has been clarified. The text now reads “Descriptive statistics are presented in the text as well as the results of chi square tests to explore gender differences.”

- Methods for how the quantitative and qualitative analyses were conducted should be detailed in the Methods section for each group of participants identified (i.e, student survey, staff survey, focus group). For the qualitative analyses, it would also be helpful to know how themes were identified, who reviewed qualitative responses, and if themes were triangulated with other sources (e.g., observations during focus groups, etc).

Descriptions of the analysis for qualitative (Stage 1) and quantitative (stage 2) analysis have now been provided. For the quantitative analysis, the text now reads “Data were analysed using STATA. Descriptive statistics are presented in the text as well as the results of chi square tests to explore gender differences.”

For the qualitative analysis, the text reads “Qualitative data were transcribed and thematic analysis conducted by two members of the research team (HY and CT). The findings that emerged from the focus groups and interviews were analysed together and identified the candidate components to take forward into stage 2, around which the questionnaires were formed.” We would like to clarify that the data from the interviews and focus groups were analysed together (i.e. not triangulated with other data), but that these were used to inform the next stage of the research.

Results:

- Pg 6, Line 46-47: The authors identified these quotes as coming from “School”. Does this mean Charity or is this another FE school? What is the difference between “School Focus Group” and “FE Focus Group”? Explanation of this somewhere in the text would help the readers where this information was collected.

We would like to clarify that this refers to the type of FE settings recruited, as detailed in the text in the methods “two ‘sixth form’ colleges attached to schools (England n=1, Wales n=1), and four large FE college campuses (England n=2, Wales n=2) with yearly intake of >1000 students.” School here refers to the sixth form colleges attached to schools. In order to avoid any confusion, we have changed reference to “school” to refer to “sixth form” instead.

- Pg. 8, Line 47, 50: Throughout the results section, some of the quotes have F? or M? Not sure what these “?” mean. In addition, some have numbers attached and some do not. These should be consistently presented.

Thank you for your comment. We have clarified where possible the identification of the quotes; at times in the focus groups it is not possible to identify the voice of the participant speaking. These are then referred to using a “?” rather than the students number. We have added a clarifying sentence in the text to reflect this. The text reads “Participants are coded numerically except for where the identification was uncertain; these are depicted using a “?”.

- Pg. 9, Line 27-31: Similar to the previous comment. Why use S? Since we know that these quotes are from Staff focus group, perhaps it would be best to present these as M/F as well.

We are keen to continue to use the S in our reporting to 1) maintain consistency in the style of reporting, and 2) to emphasise to the reader that these are staff comments, regardless of the gender of the staff.

Discussion:

- A general discussion regarding the feasibility of integrating these sexual health components to curriculum should be discussed and how these may be incorporated into current sexual health curriculum can provide more informative next steps for sexual health programs.

The four candidate components, and two components proposed for further piloting as a result of the study, are designed to be used as a multi-component intervention, not as part of the curriculum. One of the key findings of the study was that “Although 50-60% of students wanted to be taught about issues relating to sexual health and relationships in FE, the setting was overwhelmingly considered ‘too late’ for SRE delivery, and too challenging given the diversity of FE settings and students’ varied sexual health knowledge, skills and experience. This is consistent with existing literature highlighting the varying quality and quantity of SRE in schools which young people believe is currently delivered too late.⁴⁵” (page 12). Therefore we do not feel a discussion about the feasibility of integrating these components into the curriculum would be suitable in this context.

- The inclusion of a few points about how these components address health inequalities mentioned in the introduction can further justify the impact of this work.

At this point in the research process, how the components may address health inequalities has been hypothesised in the Online Appendix (Logic Model). We have acknowledged that future research is required to explore if and how the proposed components can address health inequalities. We have acknowledged this in the discussion by including new text to say “Future research is required to

explore if and how the proposed components can address health inequalities, as hypothesised in the logic model (Online Appendix).”

Reviewer: 2

Reviewer Name: Christine Markham

Institution and Country: UHealth School of Public Health, USA

Please state any competing interests or state ‘None declared’: None declared

Please leave your comments for the authors below: The authors present original findings from a mixed method multi-case formative research study to design and assess the feasibility of a safe

sex/healthy relationships intervention for further education settings in the UK. Although the study comprises a small number of sites (n = 6), it provides potentially generalizable findings to guide further intervention testing. Overall, the study appears to have been well-conducted, and the manuscript is well-written; however, some findings would benefit from clarification, and some terms would benefit from translation for a non-UK readership. The following comments are offered to strengthen the manuscript.

Methods, p.4. Address any issues in site recruitment. Only 6 sites participated in the formative research study. How many sites were approached for recruitment but declined to participate due to the potentially controversial (sexual) nature of the study? This seems to be a major question that is not addressed in the study. Overall, what is the acceptability among further education (FE) settings to address sexual health among their student population?

We thank you for this comment and would like to clarify that “All six FE settings invited to take part accepted the invitation. One setting withdrew prior to participation due to practical reasons; this setting was then replaced.” We have added these lines to the text. We believe this demonstrates acceptability among FE settings to address the sexual health of the student population.

Methods. P.4. For non-UK readers, clarify what is a ‘sexual health charity’? Is this something similar to, say, Planned Parenthood in the US? That is, a non-profit organization that provides sexual health education to other youth settings?

We have clarified the context of the sexual health charity. The text now reads “The sexual health charity was invited to participate as they are a key provider of services in local communities and educational programmes for children and young people as well as training for professionals and campaigning work across England.”

Methods. P.4. Similarly, what does ‘staff safeguarding action’ mean outside of the UK? This needs translation for non-UK readers. Also, ‘welfare staff’ – does that translate to counseling staff?

We have clarified the meaning of staff safeguarding action “perceptions of action that FE staff take when safeguarding students in relation to sexual health and relationships” and have also clarified the meaning of welfare staff “welfare staff (i.e. staff employed to deal explicitly with students’ health and wellbeing at FE)”

Results, Stage 2, page 8. It seems that multiple staff brought up issues of sustainability (cost, available funding) for on-site sexual health care services, and for administrative support in FE settings. How widely voiced were these concerns across staff focus groups?

These concerns were raised by three of the FE settings.

In Stages 2 & 3, it seems that these staff concerns were overlooked or not addressed again, when they seem as if they could be major feasibility barriers to the implementation of on-site sexual health and relationship services.

If funds for on-site sexual health services were sought from FE settings, this would indeed be a barrier to implementation, however funding for onsite sexual health services provided in FE settings would be from staff externally funded, for example from the National Health Service or third sector charity organisations. With regards college support for the intervention; overall staff were positive towards the intervention components taken forward but raised concerns about wider staff buy in. The concerns about ‘buy-in’ from staff are not unique to FE, nor to an intervention of this nature. For example, Adam Fletcher’s work on smoking cessation in FE settings (Fletcher et al., 2017). All interventions and implementation is dependent on a level of buy in from the college staff. We have added the text “Concerns about staff support for the delivery of services were not unique to FE, nor to an intervention of this nature.44” to clarify.

Results, Stage 2. Clarify, what was the response rate for student surveys, and for staff surveys. Again, it would be good to clarify in Stage 1, what was the overall response rate for invited sites that agreed to participate, to allow readers to assess the generalizability of these results across typical FE settings.

As outlined in the discussion, "FE settings were not always able to provide the numbers of enrolled students, and when provided, the numbers do not reflect attendees on site. This prevents the calculation of an accurate response rate or sampling frame." We have clarified the response rates for invited sites in the study in the methods section "All six FE settings invited to take part accepted the invitation. One setting withdrew prior to participation due to practical reasons; this setting was then replaced." We have added these lines to the text.

Results, Stage 3. Clarify why Sex and relationships education (SRE) was dropped at this stage, when findings from the Stage 2 student surveys indicated that ~50-60% of students would use these types of services.

Despite 50-60% of students wanting to be taught about issues relating to sexual health and relationships in FE, the delivery of sex and relationships education in an FE setting was overwhelmingly considered 'too late' for SRE delivery. The setting was also considered too challenging given the diversity of FE settings and students' varied sexual health knowledge, skills and experience. This is consistent with existing international literature highlighting the varying quality and quantity of SRE in secondary schools which young people believe is currently delivered too late. Therefore this is consistent with the notion that delivery in FE is too late.

Conversely, clarify why 'on-site sexual health and relationship services' were considered feasible and sustainable for further consideration, when staff input in Stage 1 questioned their sustainability and administrative support. Of all the potential intervention strategies proposed, this category seems the most unsustainable and potentially politically fraught, at least in non-UK settings. Clarify why the advisory group gave this strategy the "green light". What economic and political resources would need to be in place across the UK to make this a viable strategy? How feasible are these resources to secure across the board?

The advisory group reviewed the evidence presented to them on sustainability of onsite sexual health services, as well as staff buy-in and considered this a component to carry forward because within the UK context, on-site service provision is provided by re-locating the National Health Service, or charity staff and do not have a cost implication for the college. We do however concede that in other countries, particularly those without a universal health system, (free at point of contact), these on-site services may be more politically controversial and economically sustainable. We have acknowledged in the limitations that "Similarly, the work was conducted in the UK, therefore the provision of sexual health services may differ to other international contexts."

Discussion. Clarify why strategy 4 Sex and relationships education was demoted in final deliberations, when it seemed to rate fairly high in Stage 2 student surveys (~50-60+% of students endorsed specific topics).

Despite 50-60% of students wanting to be taught about issues relating to sexual health and relationships in FE, the delivery of sex and relationships education in an FE setting was overwhelmingly considered 'too late' for SRE delivery. The setting was also considered too challenging given the diversity of FE settings and students' varied sexual health knowledge, skills and experience. This is consistent with existing literature highlighting the varying quality and quantity of SRE in schools which young people believe is currently delivered too late.

Clarify, what weight did discussion with the adult and youth advisory boards have over Stage 2 constituent survey findings? Why so?

The youth advisory group of 14-25-year-olds was consulted prior to the funding application to assist in the development of the project aims and research questions. They were also consulted about the content and format of qualitative and quantitative research components, and the final intervention design. The youth advisory group were also consulted to provide explanations for the findings. The stakeholder advisory group were also consulted at each stage of the research. To clarify “the advisory groups reviewed the findings and provided contextual explanation for the results”. This sentence has been added to the paper for confirmation.

Figure 1. Include response rates for student and staff survey participation.

As outlined in the discussion, “FE settings were not always able to provide the numbers of enrolled students, and when provided, the numbers do not reflect attendees on site. This prevents the calculation of an accurate response rate or sampling frame.”

Table 1, Stage 3. Did the stakeholder consultation address resource/administrative support issues for On-site access to sexual health services that were raised in the Stage 1 staff focus groups & interviews? This seems to be a potentially major oversight that is not addressed in the narrative.

As outlined in the methods section “Breakout discussion groups aimed to finalise the intervention design and explore how to involve stakeholders in the co-production and delivery of an intervention, the content and delivery of safeguarding training, methodological approaches to data collection in FE settings, and developing sex positive FE settings.” We would like to clarify that the co-production and delivery of the intervention included the requirement of staff buy-in for the implementation of the intervention. The concerns about ‘buy-in’ from staff are not unique to FE, nor to an intervention of this nature (e.g. Fletcher et al., 2017). All interventions and implementation is dependent on a level of buy in from the college staff. We have added the following sentence into the discussion to address this “Concerns about staff support for the delivery of services were not unique to FE, nor to an intervention of this nature.44”

Table 1, Stage 3, sex and relationship education. What evidence is provided from stakeholders to support the premise that SRE delivered at the FE level is “too late” given that the majority of student survey respondents in Stage 2 endorsed this strategy?

We believe that students’ responses about wanting to learn about the topics reflects the lack of, and varied provision of sex and relationships education across the UK. This is consistent with existing literature highlighting the varying quality and quantity of SRE in schools which young people believe is currently delivered too late. (Pound et al., 2016). This notion was supported by sexual health charity

staff (leading providers of SRE in the UK) and sex and relationships experts in our stakeholder panel who deliver SRE across Wales.

Pound P, Langford R, Campbell R. What do young people think about their school-based sex and relationship education? A qualitative synthesis of young people’s views and experienced. *BMJ Open* 2016;6: e011329

Reviewer: 3

Reviewer Name: Adela Montero

Institution and Country: Associate professor, Center for Reproductive Medicine and Integral Development of Adolescents, Faculty of Medicine, University of Chile

Please state any competing interests or state ‘None declared’: None declared.

Please leave your comments for the authors below Congratulations for this research that addresses very important issues related to the sexual and reproductive health of adolescents and young people. I would like to comment that several of their findings have also been evidenced in my country, for example, the existing taboo regarding sexuality. In Chile, confidentiality to access sexual health services is very important for adolescents and young people as well as the need to have trained, empathic professionals, committed to the well-being of adolescents and young people.

Thank you very much for this great work.

Thank you for your feedback

VERSION 2 – REVIEW

REVIEWER	Christine Markham Associate Professor, University of Texas Health Science Center at Houston School of Public Health, USA
REVIEW RETURNED	17-Mar-2019

GENERAL COMMENTS	The authors have addressed the reviewers' comments comprehensively, and have revised the manuscript and tables accordingly. I have no further concerns or suggestions.
--